# Semantic Segmentation of Terrestrial Laser Scans of Railway Catenary Arches: A Use Case Perspective

**DOI:** 10.3390/s23010222

**Published:** 2022-12-26

**Authors:** Bram Ton, Faizan Ahmed, Jeroen Linssen

**Affiliations:** 1Ambient Intelligence, Saxion University of Applied Sciences, 7513 AB Enschede, The Netherlands; 2Formal Methods and Tools, University of Twente, 7522 NB Enschede, The Netherlands

**Keywords:** semantic segmentation, point cloud, railway infrastructure, deep learning, terrestrial laser scanner, catenary arch

## Abstract

Having access to accurate and recent *digital twins* of infrastructure assets benefits the renovation, maintenance, condition monitoring, and construction planning of infrastructural projects. There are many cases where such a digital twin does not yet exist, such as for legacy structures. In order to create such a digital twin, a mobile laser scanner can be used to capture the geometric representation of the structure. With the aid of *semantic segmentation*, the scene can be decomposed into different object classes. This decomposition can then be used to retrieve cad models from a cad library to create an accurate digital twin. This study explores three deep-learning-based models for semantic segmentation of point clouds in a practical real-world setting: PointNet++, SuperPoint Graph, and Point Transformer. This study focuses on the use case of catenary arches of the Dutch railway system in collaboration with Strukton Rail, a major contractor for rail projects. A challenging, varied, high-resolution, and annotated dataset for evaluating point cloud segmentation models in railway settings is presented. The dataset contains 14 individually labelled classes and is the first of its kind to be made publicly available. A modified PointNet++ model achieved the best mean class Intersection over Union (IoU) of 71% for the semantic segmentation task on this new, diverse, and challenging dataset.

## 1. Introduction

Renovation, maintenance, condition monitoring, and construction of infrastructural projects demand assessments of the current situation [1]. These processes are necessary for the evaluation of the existing situation, possibly leading to advice for re-designing aspects such as structural integrity, optimisation of traffic flow, and safety. In addition, the introduction of BIM (*Building Information Modelling*) and 3D design in general have created an increased need for accurate, up-to-date, 3D information of existing infrastructure.

Three-dimensional information can easily become outdated as the actual constructed infrastructure can deviate from the original design plans or can currently exist in an altered state [2]. Furthermore, blueprints do not always exist with a sufficient level of detail, are not available in a digital format, or only exist in 2D. These factors underline the need for accurate, up-to-date, 3D information.

At present, assessments and the subsequent translation to 3D are performed mostly manually, which is a time-consuming and error-prone task. This has given rise to technology aimed at automating the digitisation of infrastructure, such as photogrammetry and mobile laser scanning [3]. Laser scanning is a method that provides immediate 3D geometric information without any elaborate processing, which is in contrast with photogrammetry; further, accuracy-wise, laser scanning has better performance over photogrammetry [4]. Another benefit of laser scanning is its independence of illumination, which means specialised measuring trains can operate at night when the utilisation of the rail network is lower. A downside of laser scanning is the high cost of measurement devices compared to vision-based systems and the unstructured nature of the data. We believe the benefits of laser scanning outweigh its downsides; therefore, this technique was chosen for this application.

This paper evaluates several state-of-the-art approaches to semantic segmentation for the digitisation of infrastructure through a use case of railway catenary arches in the Netherlands. Catenary arches are the supporting structures above the railway track that carry the power lines for the trains, see Figure 1. The catenary system in the Netherlands consists of a variety of new and legacy arches, with custom and standardised components being mixed. Digitising the physical arches into their 3D, digital counterparts is an ongoing task. As part of this undertaking, mobile laser scans have been made of a small piece of railway track in the Netherlands (see Section 3).

Supervised semantical segmentation of the scanned scene provides a starting point for digitisation. In turn, these segments can be matched to individual components from a CAD library to create a full digital twin of the scene. This paper considers three state-of-the-art approaches, which semantically segment a dataset of catenary arches in the Netherlands, and compares their efficacy. Specifically, we address variations on PointNet++ [5], an implementation of SuperPoint Graph [6], and Point Transformer [7]. PointNet++ is considered as it has been a milestone in applying deep learning to point clouds. A catenary arch can be seen as a set of geometrically placed objects, which fits well with the objective of SuperPoint Graph to encode geometrical relations. The third method chosen is Point Transformer, as this was the first method to break the 70% mean Intersection over Union (mIoU) threshold on the S3DIS dataset [8]. Finally, implementation availability of code was taken into account when selecting these methods.

The remainder of this article is organised as follows. First, existing work on semantic segmentation in point clouds is described (Section 2). This is followed by a description of the dataset (Section 3). Next, the methodology for comparing the semantic segmentation techniques is described (Section 4). Thereafter, the results and discussions (Section 5) are provided. The discussion contains valuable pointers for future work and highlights the importance of explainable artificial intelligence. Finally, the conclusion and outlook for future work (Section 6) complete the article.

## 2. Related Work

Point clouds have vast applications in different areas of science and engineering such as the construction industry [9], digital photogrammetry [10], surveying [11], and robotics [12]. Therefore, many survey papers [13,14,15,16,17,18] have been written to compare different point-cloud-based machine learning models both technically and empirically.

Liu et al. [16] compared various deep-learning-based algorithms for different point cloud tasks, such as classification, segmentation, and object detection. The algorithms were divided into two categories, namely, raw point-cloud-based methods and tree-based deep learning. The raw point-cloud-based methods use the points directly as an input for training a deep learning model. The tree-based algorithm first forms a *k*-dimensional tree [19] (or *k*d-tree in short) representation of the raw point clouds. Local and global cues imposed by this tree structure can be exploited to progressively learn representation vectors [20]. An extensive empirical comparison of the performance of these models for a large number of benchmark datasets was also reported [16]. However, these datasets do not include railway catenary systems.

In a recent paper, Guo et al. [15] provided a comprehensive survey of deep learning methods for different point cloud tasks. The methods were categorised according to the three tasks associated with point clouds, namely, shape classification, object detection and tracking, and segmentation. The methods for each of these classes were further classified into different categories such as projection-based, point-based, object detection, object tracking, scene flow estimation, semantic segmentation, instance segmentation, and part segmentation ([15], Figure 1). Guo et al. not only briefly described the datasets but also commented on evaluation metrics. Furthermore, a chronological overview of the methods for all three categories was also given. The algorithms are empirically compared via different standardised metrics using benchmark datasets.

Although the surveys described above are comprehensive in describing the algorithmic advancement concerning various tasks related to point cloud data, they lack the aspect of one dataset, namely, datasets related to railway infrastructure. In the remainder of this section, the literature on point clouds related to railway infrastructure is surveyed. The reader should be warned that the performance metric used by various authors is not consistent—F1-score, accuracy, and mIoU are all used.

Arastounia used a high-density point cloud covering 550 m of Austrian railroad for segmenting individual catenary components [21]. A heuristic method was employed for this task based on the local neighbourhood structure, the shape of objects, and the topological relationship between objects. Each of the objects being segmented had a different model. The first step towards segmentation was detecting the track bed, which acted as a reference base for detecting the other components such as tracks, poles, and wires. In total, six different objects were segmented, with an average accuracy of 96.4% being obtained.

Chen et al. [22] used a more data-driven approach towards segmentation of catenary arches. Their approach starts by extracting line primitives at three different scales from the point cloud data. These line primitives are then used for training a hierarchical Conditional Random Field (CRF) model. A total of ten different objects were segmented with an overall accuracy of 99.67%.

Soilán et al. compared two deep-learning-based approaches, PointNet [23] and KPConv [24], for the task of segmenting railway tunnels [25]. Four classes were defined: tunnel lining, tracks, wires, and ground. The PointNet model achieved an average F1-score of 86.7% and the KPConv approach achieved an average F1-score of 87.2%. It is surprising that no data augmentation methods were used because the number of samples is small. An additional surprise is the low F1-score on segmenting the tracks, which have a very consistent geometric shape. This low score is attributed to labelling errors by the original authors.

The works of Chen et al. and Lin et al. share the same dataset [26,27]. This dataset was collected using a mobile, 2D laser scanning device mounted on a cart moving along the railway track to collect data. The data consist of sequences of 2D slices, which are perpendicular to the direction of the railway track. The sensor location is constrained to the track, making it very easy to define a constrained search area. We hypothesise that the variation within the captured railway catenary system is small, resulting in highly accurate results for both works. Due to the sequential nature of the data, Chen et al. opted to use a *Recursive Neural Network* (RNN) [26]. First, each of the slices is partitioned into non-overlapping regions of points using an iterative point partitioning algorithm. After this, PointNet [23] is used to derive local features from these regions. Thereafter, an RNN based on *Long Short-Term Memory* (LSTM) architecture is used to segment the points. Seventeen classes of catenary components are defined for the segmentation task. Obtained accuracies in terms of mIoU were extremely high, even smaller components such as droppers and suspension insulators achieved scores of 90.8% and 97.8%, respectively. The approach of Lin et al. [27] to segment the point clouds into individual catenary components is by first classifying each of the slices into one of the following categories: wires, droppers, or poles [27]. After this, adjacent slices with the same category are grouped together. For each of the groups, a different deep-learning-based segmentation model is trained. In total, eight classes were segmented, and a mean accuracy of 97.01% was achieved.

We hypothesise that the majority of the work on semantic segmentation of catenary systems rely on data from scenes with little variation. To determine the robustness of the segmentation models, the work presented here depends on a dataset with a large variety of catenary arches. In addition, a large number of components (14) is segmented.

## 3. Catenary Arch Dataset

This section details the process of creating the catenary arch dataset. First, the acquisition of the raw data is described. Thereafter, arch localisation, cropping, and labelling are addressed. Finally, a summary of the dataset, both visual and textual, is provided.

### 3.1. Acquisition

To the best of our knowledge, there are no publicly available point cloud datasets of railway catenary arches. Therefore, our work is based on a dataset provided by Strukton Rail, containing an 800 m stretch of railway track near Delft, the Netherlands containing 15 catenary arches, which has been digitised into a point cloud. The point cloud data were collected with a Trimble TX8 laser scanner using the level 2 operation mode. This model has a scan duration of three minutes and a point spacing of 11.3 mm at 30 m. Points are referenced within the Rijksdriehoeksstelsel [28] coordinate system, a national standard coordinate system of the Netherlands.

### 3.2. Arch Localisation

The scanned stretch of railway track was made available by the data provider in four chunks of data. A semi-automated method was used to detect the location of the catenary arches within these chunks of data. This method follows a similar approach as described by Zhu et al. [29] and Corongiu et al. [30]. The method is based on the assumption that poles are represented by a dense volume in the z-direction.

Our method first downsamples each chunk of data using a voxel filter with a cell size of 10 cm. This cell size enables the detection of poles and reduces the computational load. After that, the scene is flattened to a two-dimensional grid by summing in the z-direction. The grid size used is 20 cm. This two-dimensional representation is written to disk as a greyscale image. Within these images, pole locations are clearly visible because of their high-intensity values. The procedural steps of arch localisation within a larger scene are depicted in Figure 2. Other elements such as trees or signalling posts also produce high-intensity regions in the image. Therefore, pixel coordinates of the outer catenary poles are manually selected and are used to define a rectangular crop region with a padding of 2 m around the catenary arch. The major axis of the rectangular crop coincides with the line being defined by the poles of the catenary arch. Each of the arches is cropped from the larger chunk of data and stored individually in an LAS file [31]. After this, each of the samples is manually labelled into 14 different classes. Labelling was performed by five students and one senior researcher. The classes labelled are as follows: top bar, pole, drop post, top tie, bracket, pole foundation, steady arm, contact wire, stitch wire, wheel tension device, dropper, messenger wire support, insulator, and unlabelled.

### 3.3. Data Summary

A summary of the data is provided in Table 1. It shows that the number of points in a catenary arch ranges between 1.6 and 11 M points. In total, the dataset contains roughly 55.4 M points. Not all classes are always present in each sample; for instance, tension wheels are only needed every few arches and, thus, occur less frequently in the dataset.

A graphical overview of the entire dataset is provided in Figure 3. Some arches still have the track bed present; this is due to the fact that different individuals labelled the data. Some samples had the ground removed using an approximate progressive morphological filter [32]. The overview also clearly shows the large variation of catenary arches. For example, some arches span two adjacent tracks whilst others span four adjacent tracks.

The dataset has a large imbalance in the distribution of the classes, which is inherent to the type of object, see Figure 4. The three largest classes (unlabelled, pole, and top bar) jointly constitute 72.3% of the points in the dataset. On the other hand, the three smallest classes (dropper, stitch wire, and wheel tension device) constitute only 1% of the dataset.

## 4. Methodology

Three different deep learning models are evaluated with regards to the semantic segmentation task of point clouds. The models evaluated are PointNet++, SuperPoint Graph, and PointTransformer. The first subsection describes the general pre-processing of the data, augmentation procedures, and the metric used for evaluation. The subsections following this describe the details of each of the three individual models.

### 4.1. General Pre-Processing

Point clouds collected using a mobile laser scanner have a non-uniform density where the density decreases as the distance from the laser scanner becomes larger. In pursuit of a uniform density within each sample, each of the samples is downsampled using a voxel centroid nearest neighbour filter with a cell size of 1 cm. A centroid nearest neighbour approach is used to preserve the local point density distribution within a cell.

Each of the samples is normalised by centring the scene to the midpoint of the data span. In addition, a scaling factor is used to limit the range of coordinate values between −1 and 1. Taking into consideration the maximum dimension of a catenary arch in the dataset as 24 m and adding 3 m of safety margin, the resulting maximum dimension would be 27 m. Therefore, the appropriate scaling factor to limit coordinates between −1 and 1 is set to 13.5 m (half of the maximum dimension).

To increase the robustness of the models and to artificially increase the variations of the data seen by the models, various data augmentation techniques are used. The following three augmentations are sequentially applied to the input point cloud.

Uniform random rotation between −180° and 180° of the points around the *z*-axis;Uniform random translation of the point coordinates between −1 m and 1 m in all directions;Adding random noise to the points. The random noise is selected from a truncated normal distribution with a mean of zero, a standard deviation of 2 cm, and truncated at ±5 cm.

The parameters for the additive noise are chosen based on intuition and the facts that the laser beam has a width of 10 mm at 30 m and the smallest object (insulator) for segmentation has a maximum dimension of ≈30 cm.

A summary of the processing steps described previously is provided in Figure 5.

As the number of samples of the dataset is limited, a leave-one-out cross-validation procedure is used. In total, 14 different components are segmented within the scene. The *mean Intersection over Union* (mIoU) per class is used as a performance metric. This metric weighs all the classes equally and is independent of the class size. It should be noted that not all components are always present in each sample. As an additional overall metric, the mean of the mIoU across all samples is reported.

As the dataset has a large class imbalance, each of the models is also trained with a loss function that incorporates the class weights—that is, classes that are rare will have a higher weight compared with the more common classes.

The models are trained on a desktop computer with 64 GB of RAM, an AMD Ryzen Threadripper 2950X CPU, and two NVIDIA TITAN V GPUs with 12 GB of memory each.

### 4.2. PointNet++

The data pre-processing for the PointNet++ models follows the procedure outlined in the previous section. To ensure a fixed number of points per sample, the pre-processed point cloud is randomly downsampled to 131,072 ($2^{17}$) points. This number of points still shows sufficient density for the smaller objects in the scene such as insulators, and also allows for a small batch size of four. This means the benefits of batch normalisation [33] can still be reaped. A limiting factor to the number of points sampled is the available memory on the GPU. The normalisation method described in the previous section deviates from the original PointNet++ work, which scales each individual sample to a unit circle. A fixed scaling factor is used since the model does not need to learn to be scale-independent. As a baseline, a vanilla PointNet++ [5] model is trained to perform the semantic segmentation task. After the baseline is established, the PointNet++ model is modified to enhance the segmentation of smaller objects within the scene.

This modification consists of adding an additional *set abstraction* level to the model. The parameters are set such that they are in line with the sequence of the other parameters. The number of points is set to 2048, the radius is set to 0.05, and the size of the multi-layer perceptron is set to [16,16,32].

The steps per epoch are set to 4, which ensures that the model has encountered all training samples at least once during each epoch. The training parameters are selected based on preliminary explorative research. The model is trained for 400 epochs with a learning rate of 0.01 followed by 200 epochs with a learning rate of 0.001. The choice was made to use a fixed number of epochs because with the leave-one-out approach there is no validation set, which can be used to determine an early stopping trigger.

### 4.3. SuperPoint Graph

The SuperPoint Graph (SPG) [6] method first geometrically partitions the input cloud into individual segments using an unsupervised global energy model. These segments are referred to as superpoints in the article and represent the nodes of the graph. Nodes are connected by edges that have feature attributes such as volume and surface ratios of the two nodes being connected. The features and the superpoints are then used as input to a neural network.

Calculations of some of these edge features make use of the Quickhull algorithm [34]. Due to calculation imprecision and given the complex structure of the input point clouds, the algorithm is not always able to calculate the convex hull of the point cloud. In our case, it was not possible to create the graph in five out of the fifteen arches. An envisioned solution to overcome the imprecision issue is by enabling the ‘joggle input’ option of the Quickhull tool. This option randomly perturbs the input point cloud before running the Quickhull algorithm.

Considering that only a small part of the dataset can be used, the derived results would not be representative. Together with the fact that the generation of the graphs is computationally expensive, the decision was made not to include the results of the SuperPoint Graph in this article.

### 4.4. Point Transformer

Self-attention networks [35] are a major milestone in the area of deep learning. These networks made a significant impact in the areas of computational linguistics [36] and computer vision [37]. The goal of the Point Transformer model is to apply the concept of self-attention to point clouds.

In contrast to the PointNet++ model, this model is trained on subsets of the catenary arch. A subset is created by selecting a random point within the arch together with its nearest neighbours. A *k*-d tree is used to efficiently query the nearest neighbours of a point—in this implementation, 4096. During each training step, an arch is selected at random from which to draw the subset of points. To compensate for the fact that not all arches contain the same number of points, the probability of choosing an arch is proportional to the number of points in the arch.

The training parameters are selected based on preliminary explorative research. The model is trained for 100 epochs with 100 steps per epoch. The initial learning rate is 0.1, which is decreased to 0.01 after 60 epochs and decreased once more to 0.001 after 80 epochs. A batch size of 32 is used.

## 5. Results and Discussion

The mIoU per class for both the PointNet++ model and the Point Transformer model can be seen in Table 2. The numbers in bold indicate the highest scores. The classes are ordered subjectively based on size from large to small.

It is challenging to objectively compare the performance of both models as they are trained using two different methods of feeding in the input data. The PointNet++ model trains on the downsampled version of the entire arch, whereas the Point Transformer trains on subsets of individual arches.

Overall, the modified PointNet++ model has the best performance in terms of mean class and mean sample accuracy. On the other hand, when counting the best performing metrics, the Point Transformer model is clearly superior. It performs best for eight out of the fourteen classes, with its performance entirely pulled down by the following three classes: top tie, bracket, and wheel tension device.

It is surprising that even with a small number of samples during training, good results can be obtained. This can be attributed to the fact that most of the catenary arch components have a well-defined geometrical structure, which does not vary between instances. This might also explain the fact that applying class weights to the loss function does not give a significant performance boost for the case of the PointNet++ model. In contrast, the Point Transformer model shows a large gap in performance when comparing the weighted and non-weighted results. It is unknown why this is the case.

One of the difficulties of using this dataset is the large differences in the sizes of objects, which require segmentation. For instance, an insulator measures approximately 30 cm and a top-bar might measure approximately 24 m, which is a factor of 80 difference with respect to the size of the insulator. This difficulty translates into the fact that large objects are segmented more accurately compared with smaller objects. The modified PointNet++ model shows an improvement in terms of class mIoU for the smaller components such as the droppers, messenger wire supports, and insulators.

The dataset contains an unlabelled class, which contains all points that do not fall into one of the other categories. Even though the models are able to correctly classify this class to a certain extent, it is difficult to understand how it does so. Does it learn to recognise the large variety of features associated to the unlabelled class? Or does it learn the process of elimination—that is, if a point does not belong to one of the thirteen other classes, then must it be an unlabelled point? These questions highlight the necessity of *explainable artificial intelligence* [38] to discover the hidden, underlying functionality of such models.

Exploring the explainability of a deep learning model is a challenging task on its own [39,40,41,42]. As a preliminary step, we focused on the shape and location of an object since these two aspects are most important for the semantic segmentation task. We applied transformations such as translation and rotation to the example dataset and measured the segmentation performance of the PointNet++ based deep learning model. The preliminary results indicate that shape has an almost negligible effect on the segmentation performance. Changing the object’s location significantly affects the performance (these results can be found in the student paper [43]). We are currently devising more experiments to explore the explainability and robustness of the model.

Another interesting question that arises when trying to segment a large number of classes is whether the performance degrades when the number of classes is large. Additionally, it is hypothesised that having dedicated models for each individual class is beneficial.

When comparing the per-class mIoU of both models, the wheel tension device stands out, which the PointNet++ model is still able to segment reasonably well. On the other hand, the Point Transformer model has poor performance for this class. This could be caused by the unique shape of the wheel tension device, which is circular, flat, and symmetrical. This is another example where explainability of the model can aid in the understanding.

## 6. Conclusions

This work evaluated three deep-learning-based point cloud segmentation methods (PointNet++, SuperPoint Graph, and Point Transformer) in a real-world scenario. A custom dataset containing high-resolution point cloud scans of catenary arches was collected for this application. The arches were manually labelled into 14 different classes. To the best of our knowledge, this is the first high-resolution point cloud dataset of catenary arches that is available to the public.

Overall, the modified PointNet++ model performed best, achieving an average class mIoU of 71%. However, when counting the number of best-performing metrics, the non-weighted Point Transformer is superior. Its mean class performance was dragged down by just a few classes. The SuperPoint Graph model was not deemed appropriate for this use case as it was very prone to calculation imprecision and had high computational demands.

To counter the substantial class imbalance of the dataset, the models were also trained using class weights. Surprisingly, this had a negligible effect on the result for the PointNet++ models, yet for the Point Transformer model it was destructive.

### Outlook

Semantic segmentation of railway scenes provides a crucial stepping stone towards automated condition monitoring. For instance, the work of Burton and Heuckelbach is focused on vegetation monitoring [44]. Their work uses point cloud data to assess the risk of trees falling on the railway track. With the help of semantic segmentation, the track, masts, wires, relay cabinets, and other assets can be identified and the risk of a falling tree can be evaluated per object type.

Other opportunities arise when wires are also part of the semantic segmentation process. This enables monitoring parameters such as sag and stagger [45,46] of wires.

If catenary masts are part of the segmentation process, their tilt can automatically be determined [47]. Maintenance can be planned if certain thresholds for tilt are exceeded. Measuring the same piece of track at multiple epochs leads to an even more advanced maintenance paradigm: predictive maintenance. For instance, in the case of mast tilt, it would be possible to determine a tilt velocity. This can be used to make projections in the future and can aid the creation of optimal maintenance plans.

A bottleneck for semantic segmentation in real-world scenarios is the availability of labelled data. Creating such a dataset is tedious, time-consuming, and prone to human errors as the classes are manually labelled. These datasets also tend to be inflexible. For instance, adding a new class would require another iteration of manual labelling. To address this issue, the possibility of a more model-driven approach, where models from an existing cad library are used, can be explored. For instance, Vock et al. proposed a robust method for template matching within point clouds [48]. In our case, the templates can be generated from existing cad libraries [49]. Such an approach could be feasible as components of the catenary arch have strong geometric shapes. An alternative approach is to use an active learning paradigm [50] for reducing the labelling cost. It is possible to leverage the trained model in this feat for the human-in-loop approach. This technique has been applied successfully for image classification tasks [51] and also object detection in point clouds [52]. Given the additional computational challenge, exploring its applicability for point cloud segmentation opens new possibilities for research.

The current dataset was collected using a mobile laser scanner mounted on a tripod; such a solution would not be viable when moving towards a more production-ready solution. Therefore, the segmentation models should also be evaluated on data captured by a train-mounted mobile laser scanner. This poses new challenges such as lower resolution, shadow effects, and merging data together from multiple trajectories.

To further improve the segmentation quality, the possibility of adding additional features such as colour and intensity values should be explored. The current work only considers cropped out arches, further research should focus on segmenting entire railway scenes.

## Figures and Tables

**Figure 1 sensors-23-00222-f001:**
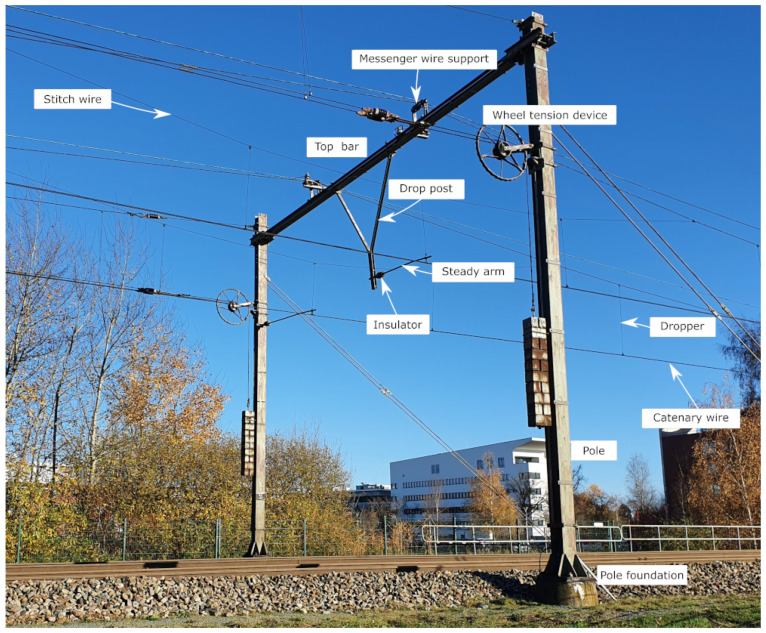
An example of a catenary arch (not in dataset) that shows the labels of the majority of the classes (own work).

**Figure 2 sensors-23-00222-f002:**
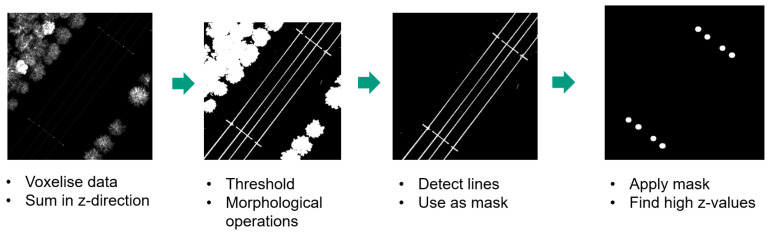
Processing steps for locating catenary arches within a large scene.

**Figure 3 sensors-23-00222-f003:**
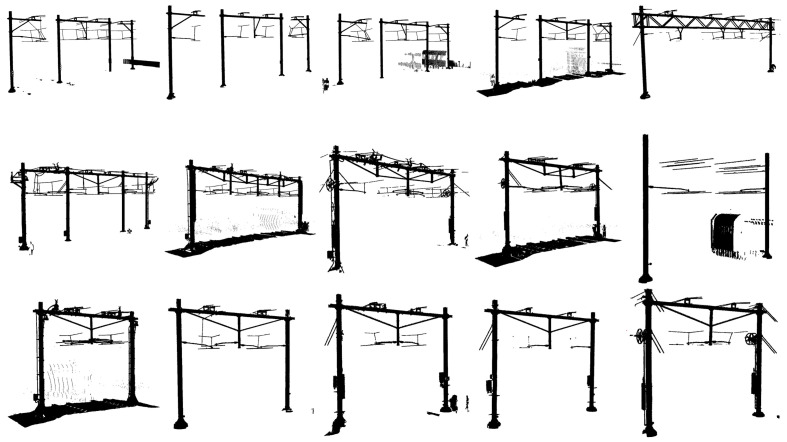
Overview of the dataset. Note the large variation of catenary arch types present in the dataset.

**Figure 4 sensors-23-00222-f004:**
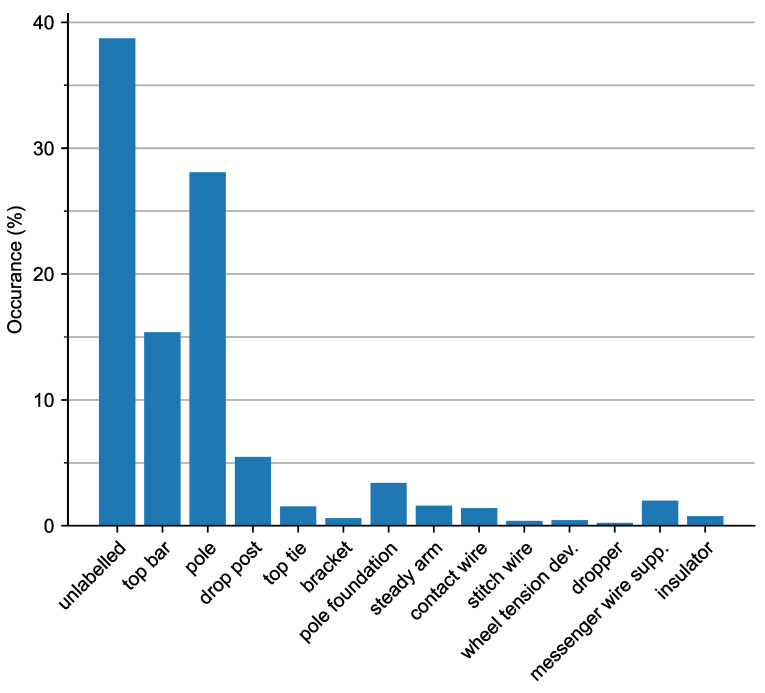
Normalised class distribution after the voxel centroid nearest neighbour filter is applied.

**Figure 5 sensors-23-00222-f005:**
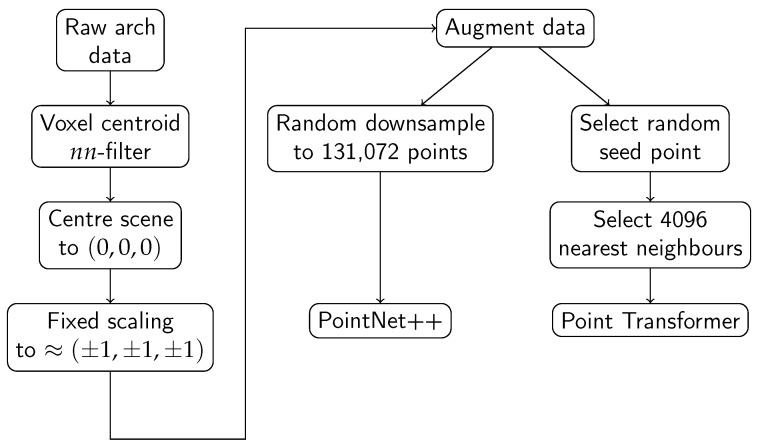
Overview of the processing steps, from the raw arch data to the data fed to the machine learning model.

**Table 1 sensors-23-00222-t001:** Statistical description of the dataset.

Arch	Name	Points	Classes (out of 14)
0	01_01	1,586,927	13
1	01_02	2,147,546	13
2	01_03	2,664,907	13
3	02_01	11,112,574	13
4	02_02	2,415,930	11
5	02_03	4,362,055	11
6	02_04	5,257,501	11
7	03_01	2,787,253	12
8	03_02	6,782,568	10
9	03_03	1,973,730	6
10	03_04	6,582,344	11
11	04_01	2,271,179	11
12	04_02	1,673,804	11
13	04_03	1,598,090	11
14	04_04	2,183,600	12

**Table 2 sensors-23-00222-t002:** Per-class mIoU compared for the vanilla PointNet++ model, the modified PointNet++ model, and the Point Transformer model (standard deviations between parentheses). The term *nw* refers to non-weighted losses and *iw* refers to inversely weighted losses.

Class	PointNet++				Point Transformer	
	Vanilla		Modified		Vanilla	
	nw	iw	nw	iw	nw	iw
unlabelled	0.63	0.63	0.69	0.67	**0.73**	0.43
top bar	0.73	0.73	**0.80**	0.78	0.78	0.70
pole	0.81	0.81	0.83	0.82	**0.89**	0.76
drop post	0.77	0.77	**0.81**	0.79	0.80	0.64
top tie	0.42	0.59	**0.83**	0.79	0.32	0.20
bracket	0.59	0.74	**0.88**	0.82	0.33	0.26
pole foundation	0.60	0.60	0.67	0.66	**0.74**	0.48
steady arm	0.54	0.54	0.58	0.58	**0.70**	0.63
contact wire	0.65	0.65	0.69	0.68	**0.71**	0.69
stitch wire	0.60	0.67	**0.71**	0.68	0.58	0.60
wheel tension device	0.52	0.44	0.70	**0.76**	0.07	0.09
dropper	0.31	0.31	0.51	0.46	**0.54**	0.39
messenger wire supp.	0.45	0.52	0.69	0.64	**0.73**	0.50
insulator	0.33	0.38	0.48	0.46	**0.76**	0.58
class mean	0.57	0.60	**0.71**	0.69	0.62	0.50
	(0.15)	(0.14)	(0.12)	(0.12)	(0.22)	(0.20)
sample mean	0.58	0.60	**0.68**	0.66	0.65	0.50
	(0.10)	(0.10)	(0.12)	(0.11)	(0.15)	(0.11)

## Data Availability

The dataset related to this article can be found at https://dx.doi.org/10.4121/17048816, an online data repository hosted at 4TU.ResearchData [53].

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
