# Peer review of "Semantic Segmentation of Terrestrial Laser Scans of Railway Catenary Arches: A Use Case Perspective"

_sensors, 2022, doi:10.3390/s23010222_

Round 1
Reviewer 1 Report
The ms focuses on a practically important issue of railway catenary monitoring, is reasonably well written, and deserves publication by Sensors. An automated monitoring would indeed provide actual information of the condition of the catenary, especially on infrastructure that have been serviced over decades by different subcontractors, including localized replacements of certain elements, e.g. wireframe poles by concrete ones or vice versa, and so on. Thus the proposed solution represents an important instrument for the infrastructure owner useful for future reconstruction planning.
In my opinion, another key goal of the monitoring would be an early detection of defects, such as inclined poles and/or gantries (steady arms), abnormally curves catenary segments, hanging wires etc. It would be nice if the authors comment on the potential ability of their approach for an eaerly detection of typical catenary defects, and provide some possible outlook.
Author Response
Dear reviewer,
First of all we would like to thank you for your time reviewing this article and the comments made. They have been valuable in the process of improving our manuscript. We have addressed the issues raised, please find our remarks, indicated by a '>', below.
In addition the exact changes compared to the original manuscript can be found in the attached document.
Kind regards,
Bram Ton
-----------------------
In my opinion, another key goal of the monitoring would be an early detection of defects, such as inclined poles and/or gantries (steady arms), abnormally curves catenary segments, hanging wires etc. It would be nice if the authors comment on the potential ability of their approach for an eaerly detection of typical catenary defects, and provide some possible outlook.
> The conclusion section has been expanded with an 'Outlook' section. This section now comments on the potential implications of our work and provides an outlook.

Reviewer 2 Report
The paper entlited "Semantic Segmentation of Terrestrial Laser Scans of Railway Catenary Arches: A Use Case Perspective" shows and interesting publication about the application of DL methods for the segmentation of railway point clouds.
Overall the paper is well structures, but some minor changes need to be included before publication:
1/Author must include in the introduction some clarifications about the use of mobile mapping for infrastructure mapping.
2/VoxeliseàVoselize in figure 2
3/Line 210: authors propose the use of data augmentation techniques in order to improve the accuracy of the model. Howerver it is not clear why they chose a normal distribution with a standar deviation of 2 cm . Plase clarify this aspect because is important for the understanding of the methodology.
4/ From my point fo view there is not sense about point 6.1 since is out of the scope of the paper. My suggestion is to delete it and focus on the conclusions of these work.
Author Response
Dear Reviewer,
First of all we would like to thank you for your time reviewing this article and the comments made. They have been valuable in the process of improving our manuscript. We have addressed the issues raised, please find our remarks, indicated by a '>', below.
In addition the exact changes compared to the original manuscript can be found in the attached document.
Kind regards,
Bram Ton
-----------------------
1/Author must include in the introduction some clarifications about the use of mobile mapping for infrastructure mapping.
> The choice for mobile laser scanning has now been better motivated in the introduction section.
2/VoxeliseàVoselize in figure 2
> The red wavy line has been removed from the figure. The British English language has been used for the manuscript, hence the *ise instead of the more American English *ize.
3/Line 210: authors propose the use of data augmentation techniques in order to improve the accuracy of the model. Howerver it is not clear why they chose a normal distribution with a standar deviation of 2 cm . Plase clarify this aspect because is important for the understanding of the methodology.
> The choice for this value has been better motivated.
4/ From my point fo view there is not sense about point 6.1 since is out of the scope of the paper. My suggestion is to delete it and focus on the conclusions of these work.
> We agree, this work is still very premature and does not fit the scope of the paper. We have therefore removed it. Furthermore the conclusion has been extended with an outlook for future work.
